# Diaza-1,3-butadienes as Useful Intermediate in Heterocycles Synthesis

**DOI:** 10.3390/molecules27196708

**Published:** 2022-10-09

**Authors:** Jorge Heredia-Moya, Daniel A. Zurita, José Eduardo Cadena-Cruz, Christian D. Alcívar-León

**Affiliations:** 1Center for Biomedical Research (CENBIO), Eugenio Espejo College of Health Sciences, Universidad UTE, Quito 170527, Ecuador; 2Facultad de Ciencias Químicas, Universidad Central del Ecuador, Francisco Viteri s/n y Gilberto Gato Sobral, Quito 170521, Ecuador

**Keywords:** diaza-1,3-butadienes, heterocycles, synthesis

## Abstract

Many heterocyclic compounds can be synthetized using diaza-1,3-butadienes (DADs) as key structural precursors. Isolated and in situ diaza-1,3-butadienes, produced from their respective precursors (typically imines and hydrazones) under a variety of conditions, can both react with a wide range of substrates in many kinds of reactions. Most of these reactions discussed here include nucleophilic additions, Michael-type reactions, cycloadditions, Diels–Alder, inverse electron demand Diels–Alder, and aza-Diels–Alder reactions. This review focuses on the reports during the last 10 years employing 1,2-diaza-, 1,3-diaza-, 2,3-diaza-, and 1,4-diaza-1,3-butadienes as intermediates to synthesize heterocycles such as indole, pyrazole, 1,2,3-triazole, imidazoline, pyrimidinone, pyrazoline, -lactam, and imidazolidine, among others. Fused heterocycles, such as quinazoline, isoquinoline, and dihydroquinoxaline derivatives, are also included in the review.

## 1. Introduction

Diaza-1,3-butadienes have recently emerged as useful organic synthesis intermediates for the construction of heterocyclic compounds. Diaza-1,3-butadienes are 1,3-butadienes having two nitrogen atoms in their structure, and they comprise 1,2-diazadienes **1**, 1,3-diazadienes **2**, 2,3-diazadienes **3**, and 1,4-diazadienes 4 (Figure 1). All diazadienes, with the exception of **1**, include imino moieties; nevertheless, because 1,2-diaza-1,3-butadienes **1** are synthesized from hydrazones or α-halogenated-hydrazones, they will also be analyzed in this review.

The potential of some diazadienes to participate in cycloaddition reactions is particularly remarkable. In this context, 1,2-diaza- and 1,4-diaza-1,3-butadienes have received more attention than 1,3-diaza- and 2,3-diaza-1,3-butadiene. According to reports, the four diazadienes may participate in a variety of cycloadditions, including [4+2] [1,2], [3+2] [3], [4+1] [4] and 1,3-cycloaddition [5]. There are also reports on [4+2] and [2+2] cycloadditions in the reaction of ketenes with 1,3-diaza-1,3-butadienes **2** [6]. Because of the reactivity of these diazadienes, numerous varieties of heterocycles can be obtained [7,8].

However, the majority of recent reviews in the literature focus on the use of related compounds, azadienes, as starting materials for the synthesis of heterocycles, whereas diazadienes have gotten very little attention [9,10,11,12].

1,2-Diaza-1,3-butadienes **1**, also known as azoalkenes, can participate in conjugate additions as well as hetero-Diels–Alder reactions [1]. The dehydrohalogenation of α-halohydrazones is the most common method for synthesizing these diazadienes, although they can also be prepared by oxidizing hydrazones with TEMPO, I_2_, or HgO, or by pyrolysis of 1,2,3-thiadiazole dioxides, oxadiazinones, or 3-hydroxy-2-arylhydrazonoalkanoic acid derivatives. Due to the high instability of 4-unsubstituted electron-deficient azoalkenes, they are often prepared in situ, nevertheless, 1,3,4-substituted azoalkenes are sometimes stable enough to be isolated [13].

One of the conventional ways of synthesizing 1,3-diaza-1,3-butadienes **2** is to use *N*-substituted benzamidines; nevertheless, these compounds have received little attention, mainly because of the intrinsic instability of the simplest members of this class of compounds. However, evidence of its use, particularly in cycloaddition reactions, demonstrates that this assumption is erroneous [14]. Recent advances in the use of these dienes show their great utility as intermediates in the synthesis of diverse heterocycles; nonetheless, one of the issues with using 1,3-diaza dienes is that if they have a substitution in position 1, they cannot produce aromatic heterocycles [15]. However, 1*H*-1,3-diaza-1,3-butadienes may be synthesized using different methods, allowing this issue to be solved [14].

The 2,3-diaza-1,3-butadienes **3**, also known as azines, on the other hand, have gained attention because of their interesting molecular characteristics. These compounds can be used not only to synthesize heterocycles but also as liquid crystals [16]. Due to the presence of an N-N bond, the two imine bonds that form the azine moiety can be seen as polar acceptor groups that are oriented in opposing directions [17]. The traditional method for preparing azines involves the condensation of hydrazine with two moles of aldehydes or ketones in a refluxing environment. As a result, depending on the type of carbonyls employed, it is possible to produce both symmetrical and unsymmetrical azines [18].

The 1,4-diaza-1,3-butadienes **4** or α-diimines, which have two imine groups, are synthesized primarily through the condensation reaction of 1,2-diketones, ketoaldehydes, or glyoxal with primary amines [19]. This technique yields symmetric diimines, and despite its simplicity, preparing unsymmetrical α-diimines is difficult; however, many methods for doing so have been reported [20]. On the other hand, because of the presence of nitrogen atoms in the structure of 1,4-diazadienes, they are known as ligands in coordination chemistry since the pioneering work of Dieck et al. [21]. The 1,4-diazabutadienes are widely used as ligands in coordination chemistry and catalysis, and there are many up-to-date references on the subject [19,22,23]. The versatile ligands have been used to form coordination complexes with most of the metals in the periodic table, including heavy alkaline earth metals [24], lanthanides, and actinides [25,26,27]. However, this point will not be addressed in this review.

The current review (see Abbreviations) will present recent advances in the heterocyclic synthesis that use diaza-1,3-dienes as key intermediates. Synthetic methodologies are divided in order according to the type of heterocycle obtained.

## 2. Synthesis of Heterocycles from 1,2-Diaza-1,3-dienes

### 2.1. Five-Member Heterocycles

#### 2.1.1. Synthesis of 2-Pyrroline

Tetrahydroberberine alkaloids derivatives **5** were synthesized using a formal [3+2] cycloaddition that involved a Michael-type addition of an enamine to a 1,2-diaza-1,3-dienes followed by nitrogen cyclization as a possible mechanism. The enamine moiety of 7,8-dihydroberberine **6** attacks a 1,2-diaza-1,3-diene **7** intermediate, resulting in non-isolable zwitterionic hydrazones **8**. The formation of the 2-pyrroline ring of the **5** is favored by intramolecular nitrogen nucleophilic attack on the iminium function due to the loss of hydrogen at the α-position of the hydrazine moiety of **8** (Figure 2) [28,29]. The reaction proceeds under mild conditions to afford the product good to excellent yields after only 15 min of reaction.

#### 2.1.2. Synthesis of 2-Arylamino 5-Hydrazono Thiophene-3-carboxylates

5-Arylamino thiophenes derivatives can be prepared by a multicomponent reaction that begins with the synthesis of 3-alkylamino-2-(carbamothioyl)but-2-enoates **9**, from n-butyl amine **10**, β-ketoesters **11** and arylisothiocyanates **12**. In the first step, the amine and the β-ketoesters react in methanol at room temperature to afford the amino ester intermediate **13** which in the second step react with arylisothiocyanates bearing an electron-donator group or a weakly electron-withdrawing group, to afford **9** in appreciable yields. The reaction of **9** with 1,2-diaza-1,3-butadienes **14** produces 2,5-dihydrothiophenes **15** with good yields (Figure 3). This synthesis can be carried out separately or in one pot, even on a gram scale [30]. Finally, after acid treatment of product **15** with Amberlyst 15*H* in a mixture of acetone/water, 5-amino thiophene-2,4-dicarboxylates **16** was obtained in good yields. 

Under the acid conditions, the ketone is released by the hydrolysis of the hydrazone moiety followed by a retro-Claisen reaction to form the thiophene ring. However, basic treatment using NaOH in THF/H_2_O affords the unexpected 2-arylamino-5-hydrazono thiophene-3 carboxylates **17** in good yields [31]. In this case, the basic medium favors the hydrolysis of the ester in position 2 of **15** followed by a decarboxylation that promotes the aromatization of the ring.

The formation of **17** under basic conditions opens the possibility of using α-halohydrazones **18** to generate in situ the diazadiene **19**. Thus, amine **10** and a slight excess of β-ketoesters **11** react in solvent-free conditions at room temperature. Then, arylisothiocyanates **12** in CH_2_Cl_2_ were added, generating **9**, which finally, reacts with 1,2-diaza-1,3-butadienes **19** generated from α-halohydrazones and potassium carbonate (Figure 3) [31]. DCM was used instead of MeOH to avoid their reaction with α-halohydrazones. It is clear that the presence of two hydrogens at C-4 of 1,2-diaza-1,3-butadienes allows the aromatization process to easily obtain 2-arylamino-5-hydrazono thiophene-3 carboxylates **17**, however, if the diene has any substituent on C-4, the final aromatization will not be feasible, and the reaction will end with the formation of 2,5-dihydrothiophene derivatives **15**.

#### 2.1.3. Synthesis of Indole Derivatives

A series of substituted indoles or polycyclic derivatives with indole moiety can be synthesized in three steps from anilines and 1,2-diaza-1,3-dienes [32]. In the first step, the aza Michael addition of anilines **20** to 1,2-diaza-1,3-dienes **21**, gives α-(*N*-arylamino)hydrazones **22** with excellent yields at room temperature without the addition of any catalyst (Figure 4). In the next step, the hydrolysis of **22** yields the respective α-(*N*-arylamino)ketones **23**. Several Lewis- and Brønsted- acids were evaluated for this reaction, and the best results were obtained using Amberlyst 15H as catalyst in a mixture of methanol-acetone as solvent at 45 °C. Under these conditions, α-(*N*-arylamino)ketone derivatives **23** are obtained with excellent yield after 12 h of reaction. Finally, the cyclization reaction of **23**, using the same catalyst, produces indole **24** with good to excellent yields. Although the same catalyst is used for the last two reactions, these cannot be performed in a single step since the catalyst concentration is a crucial parameter in the synthesis of **23**. Additionally, the first reaction requires the use of relatively low temperatures and polar solvents, while the second reaction requires high temperatures and a nonpolar solvent. These conditions allow for excellent yields of α-(*N*-arylamino)hydrazones **22** and α-(*N*-arylamino)ketones **23**.

The reaction proceeds with better yields when using secondary amines and even better if they have electron-donating groups. In addition, non-symmetric amines can be used, which allows indoles to be obtained regioselectively, mainly due to steric effects. Lastly, using cyclic amines allows the last reaction to be carried out without the need to change solvent, so the respective indoles are obtained with good yields directly from the second step. 

The 1,2-diaza-1,3-dienes used for this reaction, which mainly contains electron-withdrawing substituents, allow the synthesis of indoles bearing ester, amide, and phosphonate groups [32]. However, 1,2-diaza-1,3-dienes without electron-withdrawing substituents, such as 4-unsubstituted or 3,4-dialkyl substituted diazadienos **25**, tend to degrade and therefore cannot be prepared in advance, instead, they can be generated in situ by basic treatment of α-halo hydrazones **26**, resulting in 3-alkyl- and 3-aryl 2-unsubstituted indoles **27** in good to excellent yields (Figure 5).

#### 2.1.4. Synthesis of 1,2,3-Triazole Derivatives

The reaction between excess sodium azide and dichlorodiazadienes **28** in DMSO at room temperature affords 4-azido-1,2,3-triazoles **29** with good to excellent yields. From *o*-propargyloxy-substituted dichlorodiazadienes, subsequent thermal intramolecular cyclization gives an additional triazole cycle yield oxazocine derivatives **30** in up to 86% yield [33]. However, in the thermal cyclization of 2-pyridine derived 4-azido-1,2,3-triazoles, the elimination of molecular nitrogen promotes the cyclization of nitrene at the azine nitrogen, yielding 2*H*-[1,2,3]triazolo [4′,5′:3,4]pyrazolo[1,5-a]pyridin-5-ium-4-ides derivatives **31** in good yields [34] (Figure 6).

1,2,3-triazoles can also be synthesized at room temperature without using organic or inorganic azides by [4+1] annulation of bifunctional amino reagents **32** with 1,2-diaza-1,3-dienes **33** generated from α-halo *N*-acetyl hydrazones **34** [35]. K_2_CO_3_ was used as a base in THF for both deprotonating the amino reagents and for the generation of the 1,2-diazadiene system. The 1,4-conjugated addition of the deprotonated amino to **33** produce intermediate **35** which undergoes intermolecular cyclization, losing the -OTs group, to yield the 1,2,3-triazole derivatives **36** (Figure 7). α-Halo *N*-acetyl hydrazones with Cl or Br as leaving groups can be used, yielding the expected 1,2-diaza-1,3-diene with the same yields, however, this reaction does not proceed when protic solvents are used. On the other hand, electron-donating or electron-withdrawing substituents on the aromatic ring were tolerated, as well as groups with considerable steric hindrance. Bifunctional amino reagents with different carbamates could be used to afford their triazole derivatives with good yields.

Using the same approach, unsubstituted 1,2,3-triazoles also can be synthesized. The [4+1] annulation of primary amines with 1,2-diaza-1,3-dienes, generated in situ under basic conditions from difluoroacetaldehyde *N*-tosylhydrazones **37**, afford the desired triazole derivatives **38** [36]. Deprotonation of *N*-tosylhydrazones was successfully carried out at 40°C using NaH in EtOAc, generating the 1,2-diaza-1,3-dienes **39** with excellent yields. A series of alkyl and aryl amines react with **39**, following an 1,4-aza conjugate addition, to give the fluorated *N*-tosylhydrazone intermediate **40**. Elimination of a second molecule of HF generates intermediate **41**, which is deprotonated to produce intermediate **42**. Finally, intramolecular cyclization and release of *p*-toluenesulfonic acid provide the triazole derivatives **38** (Figure 8). 

This reaction tolerates a diversity of substituents on the primary amine and can be used with both aliphatic and aromatic amines, including optically active chiral amines, and has a high tolerance to the presence of many functional groups. This strategy provides easy access to diverse 1-substituted 1,2,3-triazoles under metal-, azide- and acetylene-free conditions. This reaction also works at 25 °C in methanol using DIPEA as a base, and this condition is used when amine substituents include alcohol or carboxylic acid moieties. One remarkable application of this methodology is the gram-scale synthesis of an antibiotic drug, PH-027. Synthesis of triazoles by this mechanism is not limited to the use of difluoro compounds since dichlorohydrazones **43** also produce 1,2-diaza-1,3-dienes **44** in the presence of DIPEA as a base in ethanol/acetonitrile. Furthermore, these diazadienes react with bisindoles **45** to generate triazole-bisindoles **46** in one step with excellent yields (Figure 9) [37].

#### 2.1.5. Synthesis of Pyrazoles

The synthesis of multisubstituted pyrazoles from the reaction of 1,2-diaza-1,3-dienes with conjugated hydrazones under acid conditions provides a new synthetic tool for preparing biologically active compounds [38]. In this reaction, 1,2-diaza-1,3-dienes **47** were generated in situ by β-protonation of α,β-unsaturated hydrazones **48**. The nucleophilic addition of a second hydrazone to the 1,2-diaza-1,3-dienes formed affords the intermediate **49**, which following intramolecular cyclization by hydrazine fragment loss generates the pyrazole derivatives **50**. Nucleophilic hydrazones can tolerate electron-withdrawing and electron-donating groups on R_1_ and R_2_ substituents (Figure 10).

#### 2.1.6. Synthesis of Pyrazolone Derivatives

1,2-diaza-1,3-diene **51** with an aryl substituent at 1 position react with propargyl alcohol (**52**) under basic conditions (4 eq K_2_CO_3_) at 60 °C to afford a mixture of 3-methyl-4-hydroxy-1-phenyl-4-(propa-1,2-dienyl)1*H*- pyrazol-5(4*H*)-one **53** and 9-methyl-7-phenyl-1-oxa-7,8-diazaspiro[4.4]nona-3,8-dien-6-one **54** with low yields. A series of other bases and solvents tried in this reaction yields complicated mixtures of products, the most optimal reaction yield was achieved using an excess of alcohol as a solvent. Using allyl alcohol (**55**) instead of **52** under the same conditions provided the corresponding 4-allyl-4-hydroxy-3-alkyl-1aryl-1*H*-pyrazol-5(4*H*)-ones **56** with better yields [39] (Figure 11). The reaction mechanism was studied by DFT calculations, observing that a Michael type addition of propargyl or allyl alcohols to 1,2-diaza-1,3-dienes, followed by cyclization and [2,3]-Wittig rearrangement yields the observed pyrazolones. 

#### 2.1.7. Synthesis of Pyrroles

Cyclodimerization of 1,2-diaza-1,3-dienes **57** catalyzed for FeCl_3_ generates a series of fully substituted symmetrical 1-aminopyrroles **58** [40]. The reaction begins with [4+2] Aza-Diels–Alder cycloaddition to produce diazenyl-tetrahydropyridazine-3,4-dicarboxylates **59** that lose N_2_ and R_1_CO_2_H to yield the intermediate **60**. The oxidation of **60** results in the formation of **61**, which loses an activated proton, triggering the internal ring closure and generating intermediate **62**. Finally, sequential ring-opening of the diaziridine nucleus by keto-enolic tautomerism yields **63** that tautomerizes to produce the stable pyrrole derivative **58** (Figure 12). A series of 1,2-diaza-1,3-dienes **57** react smoothly in THF under reflux with 10 mol % of FeCl_3_ for 30 h to yield the expected 1-aminopyrroles **58**. To ensure high yields, FeCl_3_ catalyst must be added in two portions: at the beginning of the reaction and after 6 h. Different *N*-protective groups were tolerated in 1,2-diaza-1,3-dienes, however, cyclization was sensitive to steric hindrance.

#### 2.1.8. Synthesis of Thieno [2,3-*b*] Indoles Derivatives 

2-Carboxylated thieno [2,3-*b*] indole derivatives **64** were prepared in one pot reaction from indoline 2-thiones and 1,2-diaza-1,3-dienes in methanol using Amberlyst as cycling agent [41]. Initially, the chlorohydrazone **65** is prepared by the condensation of 2-chloro-3-oxopropanoate **66** with hydrazinecarboxylate **67**, and then, the indoline-2-thione **68** and K_2_CO_3_ are added. The diene **69** is generated in situ by the carbonate present in the reaction medium and then undergoes an addition at C3 by **68** to produce the hydrazone **70**. Finally, the presence of Amberlyst in the reaction medium catalyzes the formation of the thiophene ring by attack of the indole moiety’s C3 on the hydrazine carbon of **70**, followed by aromatization facilitated by the removal of the carbazate/semicarbazide derivative (Figure 13). The reaction proceeds at room temperature, and the substituents clearly affect the reaction rate. Compared to those with strong electron-withdrawing groups, the reaction is much faster when the starting products have weak electron-withdrawing or electron-donating groups. 

### 2.2. Six-Member Heterocycles

#### 2.2.1. Synthesis of 2-Pyridone Derivatives

*N*-Substituted rhodanines **71** and two molecules of 1,2-diaza-1,3-dienes **72** have been used in an unusual multicomponent reaction under basic conditions to produce 2,3,5,6-tetrahydro-1*H*-pyrrolo[3,4-c]pyridine-1,3,6-triones **73**. This reaction begins with two successive Michael-type reactions. Initially, under the reaction conditions, a carbanion of rhodanine is formed at C5, which attacks at C-4 on the **72** to yield the intermediate **74** with excellent yields. A new carbanion formed in **74** attacks a second molecule of **72** to afford **75**. Finally, **73** was obtained following the formation of the 2-pyridone ring by cyclization of **75** after the opening of the 2-thioxothiazolidin-4-one ring and the loss of CS_2_, which was the key step in the reaction (Figure 14) [42]. This reaction could be carried out in both, step-by-step (isolating **75**) and one-pot methods, with the latter resulting in a slight increase in yield of **73**.

#### 2.2.2. Synthesis of 6-Aminopyridazine Derivatives

The reaction of dichloro-substituted 1,2-diaza-1,3-dienes **76** with 2 equivalents of malononitrile allows access to highly functionalized 6-aminopyridazine derivatives **77** (Figure 15) [43]. In the first step of this reaction, a malononitrile anion is added to C-4 of the diazadiene **76** followed by β-elimination of HCl to form a new 1,2-diazadiene **78**. A subsequent addition–elimination reaction of a second malononitrile anion forms a 1-azadiene **79**. Finally, the nucleophilic attack of the hydrazine nitrogen on the cyano group generates the heterocyclization of the molecule, with the formation of 6-aminopyridazines **77** in excellent yields. A wide range of diazadienos **76** can be used in this reaction, and both electron donor and electron-withdrawing groups can be incorporated into any of the aromatic rings. If the 1,2-diaza-1,3-dienes **76** have fluorine or chlorine atom at the *ortho* position of the aromatic ring at *N*-1, an intramolecular nucleophilic substitution reaction with the participation of the amino group afforded benzo[4,5]imidazo[1,2-b]pyridazine derivatives **80** with excellent yields. The synthesis of **80** could be performed in one or two steps with no effect on reaction yield.

Dichloro-substituted 1,2-diaza-1,3-dienes **76** were synthesized from the reaction of substituted aromatic hydrazine with benzaldehyde, followed by catalytic carbon–carbon radical coupling between the corresponding hydrazones and CCl_4_.

The reaction of dimethyl malonate and cyanoacetic esters with electron-rich and electron-poor 4,4-dichloro-1,2-diazabuta-1,3-dienes **81** in THF at 20 °C using NaH as a base afford pyridazinone derivatives **82** in high yield (Figure 16) [44]. The reaction with cyanoacetic esters is chemoselective to proceed through the attack of the ester group due to its higher electrophilicity compared to the cyano group. This reaction gives a set of pyridazinones **82** with different aryl groups at positions 1 and 3. This reaction starts with a Michael-type addition of the deprotonated ester **83** on 4,4-dichloro-1,2-diaza-1,3-butadienes **81** to afford **84**, followed by chlorine leaving through deprotonation to form the azadiene **85**. This intermediate **85** can tautomerize to form a new 1,2-diaza-1,3-butadiene **86**, which is attacked by a second equivalent of **83** followed by an elimination of the second chlorine atom to form **87**. Finally, the last intermediate **87** undergoes cyclization by nucleophilic attack of the hydrazine nitrogen to the ester moiety to afford the pyridazone derivative **82** (Figure 16).

Reaction with ethyl acetoacetate does not proceed under the above conditions; however, using K_2_CO_3_ or Cs_2_CO_3_ in DMSO instead of NaH in THF, the reaction generates product **88** in moderate yield. This reaction is more sensitive to the substituent on C3, and sterically hindered aryl resulted in the formation of **88** in trace amounts. In this reaction, the nucleophilic attack of hydrazine is chemoselective to the keto group (Figure 17).

#### 2.2.3. Synthesis of Dihydropyridazinone Derivatives

The [4+2] annulation reaction of 1,2-diaza-1,3-dienes **89** with azlactones **90** produces dihydropyridazinones **91**, in moderates to good yields [45]. Using Na_2_CO_3_ as a base, the reaction was conducted at room temperature in toluene with an N_2_ atmosphere. Under these conditions, **89** was generated in situ from *N*-protected α-halo hydrazones **92**. Azalactones can also be synthesized in situ from *N*-acyl amino acids **93** using DCC to promote cyclodehydration under the same conditions, so that the reaction can be carried out in one pot. Substrates with electron-withdrawing or electron-donating groups in diazadienes were compatible with the reaction conditions, however aliphatic substituents in the *N*-protecting group of the amino acids produced dihydropyridazinones **91** in lower yields (Figure 18). On the other hand, different substituted amino acids **93**, possessing electron-withdrawing or electron-donating groups, both in the phenyl ring (R_3_) and the *N*-protecting group (R_4_) of the amino acids are well tolerated in this one-pot reaction, yielding products **91** from moderate to excellent.

1,2-Diaza-1,3-dienes **94** react with NHC-bound enolates **95**, which are formed in situ by the reaction of α-chloro aliphatic aldehydes **96** with *N*-heterocyclic carbene (NHC), through an asymmetric [4+2] annulation, yielding chiral 4,5-dihydropyridazin-3(2*H*)-ones **97** with good yields and excellent enantioselectivities. The reaction proceeds under moderate conditions and is suitable for gram-scale synthesis (Figure 19) [46].

#### 2.2.4. Synthesis of Spiro[pyridazine] Derivatives

Ketohydrazones **98**, formed by the condensation of acetophenones/tetralone/cyclohexanones and arylhydrazines, provide in situ 1,2-diaza-1,3-dienes **99** by oxidative dehydrogenation using TEMPO in excess in N_2_ atmosphere at 80 °C. This diene reacts with 3-methyleneoxindoles **100** [47] or 2-arylidene-1,3-indanediones **101** [48], through [4+2] cycloaddition reactions to provide spiro[3,3′-pyridazines] derivatives **102** and spiro[indene-2,3′-pyridazines] **103**, respectively, in moderate to good yields. Because the reaction was a concerted cycloaddition, the relative configuration of the substituents in the oxindole moiety was conserved (Figure 20).

The reaction of 3-phenacylideneoxindoles/3-aryliminooxindol-2-ones **104** with 1,2-diaza-1,3-dienes formed in situ by dehydrohalogenation of α-halogenated *N*-acylhydrazones **105** with Na_2_CO_3_ at room temperature is another method for synthesizing this class of heterocycle. Spiro[indoline-3,3′-pyridazin]-2-ones/spiro[indoline-3,3′-[1,2,4]triazin]-2-ones **106** are obtained in good yields and with high diastereoselectivity under these conditions (Figure 21) [49]. It is noteworthy that in both cases, aza-Diels-Alder reactions occur in electron-deficient dienophiles and electron-deficient dienes, and both chloro- and bromo-substituted *N*-acylhyl- or *N*-benzoylhydrazones could be used in the reaction. The effect of the substituent on the 3-phenacyldenoxindoles on the yields was marginal.

#### 2.2.5. Synthesis of Hydropyridazines

Through a sequential [4+2] and [1+2] annulation, dienes formed in situ from the basic treatment of α-halo *N*-acetyl hydrazones **107** react with crotonate-derived sulfur ylides **108** to give tetrahydropyridazines **109** [50]. The sulfonium deprotonates in the process to produce an allylic ylide, which interacts with **107** through Michael addition to produce the intermediate **110**. This intermediate undergoes an intramolecular nucleophilic addition to make ylide **111**, which then undergoes an intramolecular S_N_2 nucleophilic substitution to form the bicyclo **109** following a proton transfer (Figure 22). The yields of pyridazine products are similar for electron-withdrawing and electron-donating substituents on dienes, although the reaction is sensitive to steric hindrance at the N-1 position on **107** and at the *ortho* position on aromatic substituents.

Dihydropyridazines **112** were synthesized by [4+2] cycloaddition of enaminones **113** with in situ produced 1,2-diaza-1,3-dienes **114** in DMSO without any base at 80 °C by the reaction of I_2_ with the corresponding *N*-tosylhydrazones. The diazadienes were formed by removing HI from the α-iodo *N*-sulfonylhydrazones obtained in the first step of the reaction. The [4+2] cycloaddition of **114** with **113** is then performed to produce dihydropyridazine heterocycles **112** through amine elimination. When the reaction was carried out at 120 °C, a tosyl group was released and pyridazines **115** were obtained (Figure 23) [51]. Electron-donating and electron-withdrawing groups were tolerated in this reaction in both substrates. Additionally, *N*-tosylhydrazones derived from alkyl, aryl, or heterocyclic ketones participate in this reaction, even if they produce sterically bulky hydrazones.

The reaction of 3-tetrazolyl-1,2-diaza-1,3-butadienes **116** with electron-rich and electron-deficient dienophiles results in an unusual reactivity pattern that allows access to a variety of functionalized derivatives. The aza-Diels–Alder reaction of methyl vinyl ketone **117** with **116**, generated in situ under basic conditions from hydrazone **118**, was recently reported; however, the expected cycloadduct was not formed; instead, **116** reacts with the methyl vinyl ketone dimer **119** present in the reaction medium to afford 3-(tetrazol-5-yl)-hexahydro-7*H*-pyrano[2,3-c]pyridazine **120** in low yield. Nevertheless, directly reacting hydrazone **118** with **119** in the presence of sodium carbonate at room temperature for 30 h afford pyrano[2,3-c] pyridazine **120** in very high yield as a single regioisomer (Figure 24) [52].

A similar cycloaddition occurs with enamides instead of enaminones. Enamides derived from acetophenone undergo an inverse electron-demand aza-Diels–Alder reaction with in situ generated 1,2-diaza-1,3-dienes from α-halo hydrazones **121** to afford 1,4,5,6-tetrahydropyridazines **122** with excellent yield (Figure 25). The reaction also works perfectly using cyclic enamide to provide structurally important fused polycyclic tetrahydronaphthalene-tetrahydropyridazines **123**. In both cases, the cycloaddition proceeds with excellent regio- and diastereoselectivity [53], and both enamide and diazadiene can have electron donor or acceptor groups that are well tolerated.

The same approach was used to synthesize tetrahydropyridazines with indole scaffolds; for this, the corresponding 1,2-diaza-1,3-dienes **124**, which were produced in situ from α-halo hydrazones **125** in MeCN at room temperature in the presence of Na_2_CO_3_, react with 3-vinylindoles **126**. Under these conditions, the inverse-electron-demand aza-Diels–Alder (aza-IEDDA) reaction occurs between the **124** and the vinyl bond of the **126**, resulting in the expected 3-(2,3,4,5-tetrahydropyridazin-3-yl)-1*H*-indoles **127** in good to excellent yields with high diastereoselectivities (>20:1 dr) (Figure 26) [54]. The electronic effect and the position of the different substituents on both the **124** and the **126** do not influence the synthesis of the highly substituted tetrahydropyridazine **127** generated by this reaction. This reaction is compatible with gram-scale synthesis and, additionally, tetrahydropyridazine **127** with outstanding diastereoselectivity and enantioselectivity may be synthesized using Cu(OTf)_2_/(S,S)-iPr-box-catalyzed IEDDA cyclization.

#### 2.2.6. Synthesis of 1,2,4-Triazine Derivatives

Formaldimines **128** and 1,2-diaza-1,3-dienes **129** can undergo [4+2] cycloaddition at room temperature to produce tetrahydro-1,2,4-triazine derivatives **130** in moderates to high yields [55]. Both formaldimines and diazadiene are formed in situ from 1,3,5-triazinanes **131** and α-halo hydrazones **132**, respectively, by the presence of K_2_CO_3_ used as base. Electron-donating groups in formaldimines moieties and electron-withdrawing substituents on 1,2-diaza-1,3-dienes showed higher reactivity in this cycloaddition (Figure 27).

#### 2.2.7. Synthesis of 1,3,4-Thiadiazines

Using K_3_PO_4_ as a base, α,β-unsaturated thioesters **133** react at room temperature with 1,2-diaza-1,3-dienes **134** through a highly regioselective inverse electron-demand aza-Diels–Alder reaction to afford 3,6-dihydro-2*H*-1,3,4-thiadiazine derivatives **135** in excellent yields [56]. A series of highly reactive 1,2-diaza-1,3-dienes **134** were synthesized in situ from α-halo hydrazones **136** under the reaction conditions. A variety of functional groups, such as esters, halogenated aromatics, and heterocycles, are well tolerated by this approach (Figure 28). In the presence of lithium aluminum hydride, **135** could be further transformed into 5,6-dihydro-4*H*-1,3,4-thiadiazines **137** in moderate to good yields. This protocol tolerates substituents with electron-withdrawing and electron-donating groups at the R_1_ and R_3_ positions.

### 2.3. Miscellaneous

Using ZnCl_2_ as a Lewis Acid catalyst, 1,2-diaza-1,3-dienes **138** can react with indoles **139** at room temperature to give cycloaddition products [57]. This reaction can follow two reaction pathways to provide two different heterocycles; tetrahydro-1*H*-pyridazino-[3,4-b]indoles **140** and tetrahydropyrrolo[2,3-b]indoles **141** (Figure 29). The product obtained depends on the substituents present mainly on the indole structure. When the substituents at R_2_ and R_3_ of **139** are hydrogen, the [4+2] cycloaddition reaction gives the fused indoline heterocycles **140** in moderate to good yields. In this reaction, six- to eight-membered cyclic 1,2-diaza-1,3-dienes were used, and a wide functional group tolerance was observed at **139**. The ring-opened [4+2] byproduct **142** was formed in some reactions as a result of a rearomatization process of product **140**, and this undesirable event appears to be the cause of the decreased [4+2] cycloaddition product yields seen in some reactions.

On the other hand, when the substituents at R_2_ and R_3_ of **139** are different from hydrogen, and the 1,2-diaza-1,3-dienes **138** are not cyclic, the reaction affords the highly crowded tetrahydropyrrolo[2,3-b]indoles **141** in good to excellent yields by a [3+2] cycloaddition reaction. Density functional theory (DFT) computational chemistry was used to investigate the mechanism of the two competing reaction pathways, which indicated a slight asynchronous concerted [4+2] cycloaddition. In contrast, [3+2] cycloadditions clearly follow a stepwise mechanism. 

## 3. Synthesis of Heterocycles Using 1,3-Diaza-1,3-butadienes

### 3.1. Five Membered Rings

#### 3.1.1. Synthesis of Imidazolines

Imidazolines are important compounds found in natural and pharmaceutical products that serve as an intermediary in the synthesis of several kinds of organic molecules [58]. These compounds can also be synthesized from 1,3-diaza-1,3-butadiene, in addition to the usual routes that allow access to these heterocycles. The reaction of 1,3-diaza-1,3-butadiene **143** with benzyldimethylsulfonium tetrafluoroborate salt **144** in the presence of MTBD (7-methyl-1,5,7- triazabicyclo (4.4.0)dec-5-ene) (**145**) produces 2-imidazoline **146** in moderate yield [59] (Figure 30). The 1,3-diaza-1,3-butadiene **143** was obtained from the reaction of amidine **147** with benzaldehyde **148** in the presence of Et_3_N and TiCl_4_ and was used directly without further purification. In the next reaction, the base favors the formation of sulfur ylide by deprotonating the sulfonium salt **144**, which makes a nucleophilic addition to **143** followed by an intramolecular nucleophilic substitution to form the imidazoline ring by releases of sulfide. The reaction only produces the *trans* isomer. Although only the synthesis of **146** is reported in the article, the reaction has the potential to obtain 2-imidazoline derivatives with different substituents on the aromatic rings.

#### 3.1.2. Synthesis of Imidazo[1,2-a]heterocycle Derivatives

Due to their enticing pharmacological properties, imidazo-heterocyclic scaffolds are recognized as drug-prejudice scaffolds. Several drug molecules are having this prevalent core, e.g., sedative Zolpidem, antiulcer drug Soraprazan, cardiotonic drug Olprinone, osteoporosis drug Minodronic acid, etc. [60]. There are several methods for preparing imidazo[1,2-a]pyridine derivatives [61]; however, the Groebke–Blackburn–Bienaymé reaction (GBB) [62] is the most commonly used. In this three-component reaction, imidazo[1,2-a]pyridine derivatives **149** are prepared by a sequential combination of an aldehyde **150** with the 2- aminopyridines **151** and isocyanide **152**. Initially, the condensation of **150** with **151** produces the 1,3-diaza-1,3-butadiene **153**. The acid catalyst activates the intermediate **153**, which then undergoes a [4+1] cycloaddition with the isocyanide **152** to yield the cycloadduct **154**. Finally, aromatization of this cycloadduct via a 1,3-H shift yields the imidazo[1,2-a]pyridine **149** as the major product (Figure 31). 

Several catalyst were used for his reaction, for example, *L*-proline [63], Yb(OTf)_3_ [64], *Candida antarctica lipase B* (CALB) enzyme [65], NH_4_Cl [66,67], K-10 clay [68], AgOAc [69], thiamine hydrochloride [70], Sc(OTf)_3_ [71,72], In(OTf)_3_ [73], InCl_3_ [74], and HClO_4_ [75,76]. Nevertheless, catalyst and solvent-free conditions have also been reported [77,78]. Although most authors work directly with isocyanide, several methodologies that avoid isolating the unstable isocyanide intermediate have recently been reported. Thus, using I_2_-PPh_3_-Et_3_N reagent [79], and triphosgene, Et_3_N [80], isocyanide is synthesized in situ from *N*-formamide **155** (Figure 32).

An advantage of this reaction is that it is not limited to the exclusive use of 2-aminopyridine, but also can be used with amines with the cyclic H_2_N-C=N structure (2-aminoazines or amidines), so this reaction allows to obtain different imidazo[1,2-a]-heterocyclic compounds (Figure 33). Among the recently reported amidines we have: 2-aminothiazoles [64,69,72,78,81], 2-aminopyrazine [72,73,81], 2-aminobenzothiazole [70,78], 2-aminopyrimidine [69,71], 5-(trifluoromethyl)-1,3,4-thiadiazol-2-amine [77] and 2-aminoquinoline [70,72].

In addition, more complex heterocycles can be obtained starting from these imidazo[1,2-a]-heterocyclics (Figure 34) [64,68,72,75].

This [4+1]-annulation reaction is also observed when β-keto sulfoxonium ylides **156** are used (Figure 35). The ylide reacts with the heterocyclic azine-aldimines **157** generated in situ, releasing DMSO and producing the dihydroimidazo[1,2-a]pyridine intermediate **158**, which tautomerizes to **159**. Finally, the dehydrogenation of **159**, catalyzed by CuCl_2_, TsOH, and DMSO, present in the reaction medium, provides the corresponding 2-aryl-3-aroyl-imidazo[1,2-a]pyridine **160** in high yields [82]. 

### 3.2. Six Membered Rings

#### 3.2.1. Synthesis of Pyrimidine Derivatives

Pyrimidine derivatives have a wide range of chemical, bioorganic, and medicinal chemistry applications. These compounds are important structural components of a wide spectrum of biologically active molecules and exhibit antimycobacterial, antitumor, antiviral, anticancer, anti-inflammatory, and antibacterial properties [83,84]. The main reaction to synthesize pyrimidine derivatives from 1,3 diazadienes is through [4+2] cycloaddition reactions. In these reactions we used 2-trichloromethyl- [85,86], and 2-trifluoromethyl-1,3-diazabutadienes [15]. These 2-trihalomethyl-1,3-diaza-1,3-butadienes **161** were prepared by the condensation of trihaloacetamidine **162** with amide acetals **163** or with chloromethaniminium salt **164** derived from *N*,*N*-dimethylbenzamide with phosphorus oxychloride (Vilsmeier–Haack reagent) (Figure 36). The principal characteristic of these 1,3-diazabutadienes is their reactivity towards electron-deficient acetylenes, and they react with dimethyl acetylenedicarboxylate (DMAD) in CH_2_Cl_2_ to give the 2-(trihalomethyl)pyrimidines **165** in high yields with a small amount of dialkylamine–DMAD adduct **166** [15,86]. The reaction gives better yields when the four-position substituent on **161** is an aromatic ring bearing both, electron-donating and electron-withdrawing groups, instead aliphatic substituent.

Under basic conditions, on the other hand, the reaction of **161** with ketene, produced in situ from enolizable acyl chlorides **167**, provides the expected 2-(trichloromethyl)pyrimidin-4-one derivatives **168** via a non-concerted [4+2] cycloaddition process. Subsequently, **168** reacts with POCl_3_ to give the corresponding pyrimidine **169** [85].

These pyrimidines can undergo substitution reactions with different nucleophiles, which increases the types of derivatives that can be obtained through this reaction. When the reaction occurs at low temperature or with a low concentration of nucleophile (5–10 equiv), an S_N_Ar2 reaction is observed in position 4 of the heterocyclic ring. In contrast, if the reaction is carried out at a high temperature or with a more significant amount of nucleophile, a substitution of the trichloromethyl group is observed (Figure 36). Furthermore, when using an excess of secondary or primary amines as nucleophiles, the formation of the corresponding amide **170** is observed, which would be obtained after the hydrolysis of the intermedia iminium salt formed after the elimination of a chlorine atom [85].

A similar [4+2] cycloaddition was reported for the reaction of 1,3-diazadienes **171** with 3-vinylindoles **172 [87]**. Chiral phosphoric acid catalyzed an asymmetric inverse-electron-demand aza-Diels–Alder reaction under mild conditions, yielding a wide range of benzothiazolopyrimidines **173** with good yields and excellent diastereo- and enantio-selectivity (Figure 37). A probable concerted reaction pathway facilitated by the dual hydrogen-bonding effect was postulated to explain the excellent enantioselectivity and specific *trans*–*trans* diastereoselectivity.

#### 3.2.2. Synthesis of Dihydropyrimidine Derivatives

The aza-Diels–Alder reaction of 4-dimethylamino-1,3-diaza- 1,3-butadiene **174** with electron-deficient olefins **175**, following the elimination of a dimethylamino group produces the corresponding dihydropyrimidines **176** (Figure 38). In the first reaction, the cycloaddition of **174** with *N*-methoxy-*N*-methylacrylamide [88] or 1,2-disubstituted ethylenes [89], in the presence of Li_2_CO_3_, affords 4-dimethylamino-2-phenyltetrahydropyrimidines **177** in moderate yield. Next, the elimination reaction of the 4-dimethylamino group by reaction with MeI produces **176** in good yield. The one-pot synthesis of **176** without isolation of the cycloadducts **177** improves the yield of dihydropirimidine formed. The *N*-protecting group of **176** could be easily removed through TFA and CH_2_Cl_2_ at room temperature to obtain *N*-unsubstituted dihydropyrimidines **178** as a mixture of tautomers. On the other hand, the cycloaddition reaction of *N*-methoxy-*N*-methylacrylamide allows the formation of 6-unsubstituted 4-dimethylamino-2-phenyltetrahydropyrimidine **177**, having the Weinreb amide at position 5. Substitution reaction of this amide group with organolithium reagents, and subsequent elimination reaction with MeI, gives 4,6-unsubstituted 5-acyl-2-phenyldihydropyrimidines **179**. The same product can be obtained by reacting **176** with organolithium or Grignard reagents [88]. Finally, reduction of **177** with DIBAL-H and elimination with MeI yields 5-formyl dihydropyrimidine **180** in a good yield, and subsequent Horner–Emmons reaction extends the dihydropyrimidine conjugation at position 5 to afford **181** in high yield.

#### 3.2.3. Synthesis of Pyrimidinone Derivatives

The [4+2] cycloaddition process of conjugated 1,3-diazabuta-1,3-dienes with suitable ketene precursors is the most successful route to functionalized pyrimidinones. This approach allows, in a single step, to fuse this heterocycle with others that have potential biological activity. Thus, 1,3-diazabuta-1,3-dienes **182** generated from amine derivatives **183** react with indoleketenes via a [4+2] cycloaddition reaction to provide [1,3,4] thiadiazole [3,2-a]pyrimidin-5-one or [1,3,4] oxadiazolo[3,2a]pyrimidin-5-one derivatives **184** (Figure 39) [90,91]. 

In the presence of phosphorus oxychloride (POCl_3_), the appropriate substituted benzoic acids **185** react with semicarbazide or thiosemicarbazide to produce, respectively, amine-1,3,4-oxadiazole and amine-1,3,4-thiadiazole derivatives **183**. Lastly, condensation of **183** with *N*,*N*-dimethylformamide dimethyl acetal (DMF-DMA) at room temperature yields the 1,3-diazabuta-1,3-diene derivatives **182**. Finally, in the presence of *p*-toluene sulphonyl chloride and triethylamine, indole acid produces in situ indole ketene, which reacts with 1,3,4-thiadiazole/oxadiazole substituted 1,3-diaza-1,4-butadienes **182** via a [4+2] cycloaddition reaction, to provide the desired pyrimidinone hybrids **184** in good yields. The evidence indicates that the [4+2] cycloaddition reaction does not occur in a concerted way, instead, the nucleophilic addition of N1 in 1,3-diazabuta-1,3-dienes **182** to the carbonyl group of ketene **186** results in a zwitterionic intermediate **187**. The keto-enolic tautomerist of the intermediate **187** generates the dipolar intermediate **188**, which yields the required products after ring closure and removal of *N*,*N*-dimethylamine (HN(CH_3_)_2_).

Using the same procedure, 5-prop-2-ynylsulfanyl-pyrimidin-4-ones **189** were synthesized, with excellent yields, by reacting 1-aryl-2-phenyl-1,3-diazabuta-1,3-dienes **190** with the acid **191**. Prop-2-ynyl-sulfanyl ketene **192** is synthesized in situ under the same reaction conditions, and it reacts with the **190** in the same way described above to produce **189** (Figure 40). Changes in the *N*-aryl moiety of the diazabutadienes **190** have almost no effect on yield. At room temperature, the reaction of **189** with iodine resulted in the simple and chemoselective synthesis of pyrimido[5,4-b][1,4]thiazin-8-ium iodide 193 in good yields. These cyclizations occurred after a favorable exo-dig intramolecular ring closure cyclization, with no evidence of the formation of pyrimidino[5,4-b][1,4]thiazepane **194** by competing *endo*-dig intramolecular ring closure cyclization [92].

Dihydropyrimidinones **195** are obtained if the diene lacks a good leaving group, as reported for the cycloaddition reaction of diazadiene **196** with ketenes synthesized in situ from enolizable acyl chloride **197** in THF in the presence of *n*-TBAHSO and powdered KOH [93]. Following this approach, the corresponding 3,7-diaryl-6,7-dihydro-5*H*-6-substituted thiazolo[3,2-a]pyrimidin-5-ones **195** can be synthesized in good to excellent yields under heat conditions or utilizing a phase transfer catalyst combined with ultrasonication (Figure 41). Initially, the diazadiene **196** is prepared when 2-amino-4-arylthiazoles **198**, previously synthesized from thiourea and bromoacetophenones, react with substituted benzaldehydes. Compared to the traditional approach, ultrasound lowered reaction rates (about 5 h) and increased yields of thiazole pyrimidinones **195**.

#### 3.2.4. Synthesis of Quinazoline Derivatives

A wide range of compounds with quinazoline or quinazolinone moieties have been found to exhibit various biological activities and may also be prepared from diazadienes. Quinazoline and quinazolinone scaffolds are types of biologically active nitrogen heterocyclic molecules that constitute the basis for several commercially available drugs [94]. Vilsmeier–Haack reagents derived from the corresponding benzamide derivatives **199** with trichloroacetamidine **200** were used to produce excellent yields of 2-(trichloromethyl)-1,3-diaza-1,3-butadienes **201.** Under mild conditions, these dienes react with benzyne, which is produced in situ from *o*-trimethylsilylphenyl triflate **202** and TBAF, to provide 2-(trichloromethyl) quinazolines **203** in good yields (Figure 42) [95]. This approach, however, is limited to *N*,*N*-substituted amides with no enolizable hydrogen atoms.

## 4. Synthesis of Heterocycles Using 2,3-Diaza-1,3-butadienes

### 4.1. Five Membered Rings

#### 4.1.1. Synthesis of Perhydro [1,2,4] Triazolo [1,2-a] [1,2,4] Triazole-1,5-dithiones

Tetrahydro-[1,2,4]-triazolo [1,2-a][1,2,4] triazole-1,5-dithione derivatives **204** can be synthesized by reacting 2,3-diaza-1,3-butadienes **205** with 2 equivalents of KSCN. This reaction has been described utilizing ultrasonic [96] or catalyzed by TiO_2_-functionalized nano-Fe_3_O_4_ encapsulated silica particles [97], with high yields, low reaction times, and simplicity of work-up in both cases (Figure 43).

#### 4.1.2. Synthesis of Pyrazoline Derivatives

Pyrazoline derivatives **206** could be easily synthesized by intramolecular cyclization of 2,3-diaza-1,3-butadienes **207** (Figure 44). Several metal salts catalyze this reaction, but FeCl_3_ has the best catalytic performance, with 95% conversion and 99% selectivity with a catalyst concentration of 4 mol%. Diene’s steric hindrance is a critical factor in the reaction since the yield of the product decreases as the size and branching of the chain increase, with yields of 10% seen in the case of *t*-Bu and no reaction observed in the case of phenyl group [98].

#### 4.1.3. Synthesis of [1,2,4]Triazolo[1,5-a]pyridine Derivatives

[1,2,4]-Triazolo[1,5-a] pyridines derivatives **208** can be easily synthesized by the reaction of 2,3-diaza-1,3-butadienes **209** and arylidenemalononitriles **210** using metallic copper as a catalyst (Figure 45) [99]. The reaction tolerates the presence of substrates with different functional groups, allowing good yields of **208**. Because of steric effects, dienes with electron-donating groups in the *para* position of the aromatic ring, afford higher yields than those with electron-withdrawing groups. Furthermore, this reaction tolerated larger aromatic groups such as naphthyl and even heterocycles. No product was obtained when TEMPO was added to the reaction, suggesting that the reaction mechanism most probably included a free radical process. However, oxygen was essential to this catalytic reaction since the desired product was only produced at a low yield under N_2_.

#### 4.1.4. Synthesis of 4,5-Disubstituted-3-amino-1,2,4-triazoles

The reaction of 2,3-diaza-1,3-butadienes **211**, previously prepared from the condensation of benzaldehyde with aminoguanidine hydrochloride, with I_2_ in dioxane at 80 degrees, provides 4-arylideneamino-5-aryl-3-amino-1,2,4-triazole derivatives **212** via intermolecular cyclization involving two molecules of **211** [100]. The iminic C4 carbon of **211** undergoes electrophilic activation by the coordination of I_2_ with the iminic nitrogen (N3), which is facilitated by the presence of AgOTf, initiating a nucleophilic attack by the N2 of another molecule of **211**, followed by the removal of HI, and formation of intermediate **213**. The terminal NH_2_ group in **213** nucleophilically displaces iodine as another molecule of HI, leading to the formation of the intermediate **214**. Finally, aromatization of the formed heterocyclic ring in **214**, following guanidine elimination and 1,3-hydrogen shift, yields the corresponding 4-arylideneamino-5-aryl-3-amino-1,2,4-triazole derivatives **212** (Figure 46). This I_2_-promoted tandem intermolecular nucleophile attack/cyclocondensation/aromatization reaction works with a wide range of aromatic ring substituents, resulting in moderate to good yields of the corresponding products. However, this reaction does not work with diazadienes derived from aliphatic aldehydes.

### 4.2. Six Membered Rings

#### 4.2.1. Synthesis of (*N*′-Substituted)-hydrazo-4-aryl-1,4-dihydropyridines

In the presence of a copper catalyst, the reaction of 2,3-diaza-1,3-butadienes **215** with alkyl propiolate **216** yielded *N*′-substituted-hydrazo-4-aryl-1,4-dihydropyridines **217** with good yields after 24 h of reaction in refluxing MeOH (Figure 47) [101]. In this reaction, the nature of the substituent on the aromatic ring of the azine has a significant effect on the synthesis of **217**. Thus, the presence of electron-donating groups favors the formation of **217**; however, there is no reaction with electron-withdrawing groups. Similarly, the presence of free hydroxyl groups in the *ortho* position does not result in the formation of a product attributable to the catalyst’s deactivation due to Cu(II) coordination with the free hydroxyls. Therefore, the reaction is limited only to **216** since when different alkynes are used, the reaction does not proceed.

#### 4.2.2. Synthesis of Isoquinoline Derivatives 

Isoquinoline derivatives **218** can be prepared by the annulation reaction of 2,3-diaza-1,3-butadienes **219** with alkynes via sequential C-H/N-N bond activation. Recently a ruthenium catalyst in PEG media with microwave radiation [102] and an air-stable cobalt catalyst in trifluoroethanol with NaOAc as an additive [103] was used in this reaction. Diazadienes **219** derived from acetophenone with electron-donating and electron-withdrawing groups at different positions of the phenyl ring were well tolerated for the reaction with various alkynes to afford the corresponding isoquinolines **218** in good to excellent yields, regardless of the catalyst used (Figure 48). Only a single isoquinoline product is detected when these substituents are in the *para* position or when the dienes have electron-withdrawing or weak electron-donating substituents in the *meta* position. However, strong electron donor groups in the *meta* position produced two regioselective isomeric products. Furthermore, diazadienes derived from other aromatic/heteroaromatic ketones proceeded through this annulation reaction, giving the expected reaction products in good yields. However, the reaction using terminal alkynes or diazadienes derived from benzaldehyde fails. The ruthenium-catalyzed process is substantially faster than the cobalt-catalyzed reaction, and microwave irradiation is required for the reaction to be effective.

## 5. Synthesis of Heterocycles Using 1,4-Diaza-1,3-butadienes

### 5.1. Four-Membered Rings

#### Synthesis of β-Lactams

The reaction of 1,4-diaza-1,3-butadiene **220** with butadienylketenes proceed via a [2+2] cycloaddition reaction to yield *cis*-butadienyl-4-iminomethyl-azetidin-2-ones **221** and butenylidene-butadienyl-[2,2′-biazetidine]-4,4′-diones **222** (Figure 49). The ratio between these products depends on the concentration of the acid chloride used to generate the ketene in situ. When the ratio is equimolar, mono β-lactams **221** are the main product, while bis-β-lactams **222** are formed with low yield. The reaction only works with aromatic 1,4-diazadienes, and the yield of **221** decreases when the reaction is carried at an elevated temperature, and no evidence of **222** is observed. On the other hand, when high equivalents of sorbyl chloride are used, an inverse behavior is observed, with the bis-β-lactams **222** as the main product, and **221** is formed in low yields [104].

When two equivalents of ketene derivatives react with unsymmetrical monophenyl cyclic 1,4-diazadienes **223**, 1,4-diazabicyclo[4.2.0]octan-8-ones **224** are obtained (Figure 50) [105]. The product is formed by combining a [2+2] cyclization to afford a β-lactam ring and a 1,5-sigmatropic rearrangement. 

### 5.2. Five-Membered Rings

#### 5.2.1. Synthesis of *N*,*N*-Diarylimidazolium Salts

One of the main heterocycles synthesized using 1,4-diaza-1,3-butadienes are imidazolium salts. These compounds are important because they are the precursors of *N*-heterocyclic carbenes (NHCs), which are generated by deprotonation [106]. The NHCs are interesting structures due to the catalytic activity of their metal complexes and are widely used in organic synthesis [107,108].

The imidazolium salts **225** could be synthesized using a mechanochemical one-pot two-step procedure, affording these NHC precursors much better yields than conventional solvent-based procedures [109]. The synthesis of diazadiene **226** and subsequent cyclization to the imidazolium salts was carried out in a planetary ball mill (pbm) at 500 rpm, with no isolation of diazadiene intermediates. Of the different reagents used for the cyclization step, the best result was obtained using paraformaldehyde and HCl (Figure 51).

Chiral imidazolium salts **227** may be synthesized in a single step with moderate yield by reacting chiral amine **228**, glyoxal, paraformaldehyde, and aqueous HCl (Figure 52) [110]. One advantage of this procedure is that it employs inexpensive starting materials, and the reaction proceeds without racemization. However, the application of this approach to construct steric hindrance imidazolium salt failed due to the too sterically bulky structure, which prevents cyclization even when the reaction is performed in two steps.

Imidazolium salt **229** can also be synthesized in one pot with 2,6-diisopropylaniline **230**, cycloalkylamines, glyoxal, and formaldehyde in HOAC, using MgSO_4_ and ZnCl_2_ as additives, and then HCl and KPF_6_ [111]. This approach yields unsymmetrical 1-(2,6-diisopropylphenyl)-3-cycloalky-imidazolium salts **229** with very high selectivity (>95%) and excellent yield (Figure 53). The reaction also works with chiral amines, allowing access to bulky chiral NHC precursors.

Reaction mechanism studies with adamantylamine **231** revealed that in the initial reaction, two symmetrical diimines (**232** and **233**) and an unsymmetrical diimine **234** can form (Figure 54). However, in additional experiments to analyze the reaction mechanism, the authors found that diaryl-diimine **232** is the main product of the initial reaction. Experimental evidence suggests that the formation of the imidazolium salt from symmetric diimines is unfavorable in this reaction and may explain the production of the asymmetric salt **235**. This might be because the cyclization of these diimines is slower than that of the asymmetric diamine **234**, as well as the rate of formation of **234**, which increases with the presence of ZnCl_2_. An alternative cyclization mechanism involving diimine **232** and hemiaminal formed by the addition of **231** to formaldehyde, on the other hand, cannot be ruled out.

Instead of glyoxal, 2,3-butanediones can be used as the starting material to make unsymmetrical imidazolium salts **236** (Figure 55). The main product of the first step in this reaction is the α-keto-imine **237**, rather than the diaryl diimine, which is obtained in low yield; however, a subsequent reaction with cycloalkylamines **238** yields the unsymmetrical imidazolium salt **236** in moderate yield, possibly following the same path as the above [112].

*N*-Heterocyclic carbene precursors with an acenaphthylene moiety **239** have a perfect combination of electronic and steric characteristics, making them particularly helpful in organic synthesis and have found broad use in the synthesis of different metal complexes. There are several reports on the synthesis of these precursors **239** [108], but in general, the initial step is the synthesis of the diimine **240**, which is accomplished by condensation of the aniline derivatives with acenaphthoquinone (**241**) dissolved in AcOH, in the presence of a Lewis acid, usually ZnCl_2_. Finally, the cyclization reaction, usually performed with chloromethyl methyl ether (MOMCl) or ethyl chloromethyl ether (EOMCl), affords the imidazolium salts **239** with excellent yields (Figure 56). This method produces imidazolium salts **239** with a broad set of substituents in the aniline moiety, which can be aliphatic or aromatic (mono-, di-, or trisubstituted) and gives access to NHC precursors with low steric congestion or even sterically more hindered.

The synthesis of imidazolium salts with higher steric hindrance is not very simple, mainly because the steric hindrance hinders the cyclization process. However, the reaction of steric diiminas with EtOCH_2_Cl for 16 h at 100 °C affords de bulky imidazolium salts **242** in high yields (Figure 57) [113].

It was recently reported that the glyoxal used in the reaction with *cis*-4-aminocyclohexane carboxylic acid not only plays an important role in the diimine synthesis but also its in situ decomposition releases carbon monoxide and generates formaldehyde, which allows cyclization of the molecule, allowing the synthesis of the zwitterionic imidazolium **243** (Figure 58) [114].

When the 1,4-diaza-1,3-butadienes **244** (R_1_ = R_2_ = H) react with trialkyl orthoformate in the presence of TMSX, heteroatom-functionalized imidazolium salts **245** are synthesized in one step. The formation of adducts between diazadiene and activated orthoformate is a crucial step in this process. The 4-sulfanylated imidazoliun salts **246** are obtained under the same conditions but with the addition of a strong RSH nucleophile. Finally, the chloroalkyl substituted imidazolium salts **247** are the main products of the reaction of **244** with TMSCl when the diazadiene possesses identical alkyl groups on the carbon atoms of the diene backbone (R_1_ = R_2_ = Me). However, if the alkyls are different (R_1_ = Et, R_2_ = Me), a combination of three imidazolium salts, both isomeric chloroalkyl substituted salts **248** and **249** and vinyl substituted salts **250**, are obtained (Figure 59) [115].

In the presence of acetic acid, the reaction of perchloric acid with 1,4-diaza-1,3-butadienes **251**, which are formed in situ by the microwave reaction of amine **252** with glyoxal trimer dehydrate **253**, yields the 2-formylimidazolium salts **254**, which are decarbonylated to imidazolium derivatives **255** in medium to high yields when heated in ethanol under reflux (Figure 60) [116].

#### 5.2.2. Synthesis of Imidazolidines Derivatives

A tandem reaction between 1,3-dialkyl-2-arylguanidines **256** and 1,4-diaza-1,3-butadienes **257** affords 4,5-bis(arylimino)-2-(alkylimino)imidazolidines **258** in moderate to excellent yields [117]. Copper(II) oxide nanoparticles catalyze the reaction between aniline derivatives and dialkyl carbodiimides to produce the initial guanidines **256**. Subsequently, in the presence of NaH, the guanidine generates the guanidinium anion **259**, which reacts with the diazadiene **257** through a nucleophilic attack to provide the iminoguanidine intermediate **260**, which then undergoes intramolecular cyclization to produce the imidazolidine derivatives **258** (Figure 61).

#### 5.2.3. Synthesis of 1,3-Thiazolidine Derivatives

When primary amines react with carbon disulfide and *N*,*N*′-diphenyloxalimidoyl dichloride **261** in the presence of Et_3_N at room temperature, 4,5-bis(phenylimino)-1,3-thiazolidine-2-thione derivatives **262** are synthesized in good yield (Figure 62) [118]. The dianion **263** is produced in the first phase of this one-pot process by reacting the amine with CS_2_. In the second step of the reaction, this dianion, a heteroanalogue of the guanidinium cation, is attacked by 1,4-diaza-1,3-butadiene **261**, affording 1,3-thiazolidine-2-thione derivatives **262**, which are formed by the elimination of 2 eq of triethylammonium chloride.

When isothiocyanates are used instead of CS_2_, 2-imino-3-aryl-4,5- bis(arylimino)thiazolidines **264** are formed (Figure 63). The reaction proceeds smoothly, and the product is synthesized in good yield. It is also proposed that a dianion heteroanalogue of guanidinium cation is formed in the first step. The dianion **265** is attacked by *N*,*N*′-diphenyloxalimidoyl dichloride **266** to form the product **264** [119].

The 4,5-bis(phenylimino)thiazolidin-2-ylidene derivatives **267** are produced by reacting acetonitrile derivative **268** with aromatic or aliphatic isothiocyanate **269**, followed by a nucleophilic reaction with *N*,*N*′-diphenyloxalimidoyl dichloride **270**. When **268** differs from malonitrile, **267** can exist as two geometrical isomers (*E*) and (*Z*). On the other hand, when the isothiocyanate derivative **269** is substituted with CS_2_, the reactions proceed smoothly and yield moderate to good yields of 4,5-bis(phenylimino)-1,3-dithiolan-2-ylidene derivatives **271** (Figure 64) [120].

#### 5.2.4. Synthesis of 1,3,2-Diazaphospholenes

1,3,2-Diazaphospholenes (DAPs) are heterocycles that contain a carbon–carbon double bond and two nitrogen atoms separated by a phosphorus atom. These compounds have been widely used as catalysts in organic synthesis, mainly due to the unique reactivity of P-hydrido substituted members of this family [121]. The most widely used protocol for their synthesis is a two-step process consisting of reduction of the 1,4-diaza-1,3-butadiene **272** with lithium or sodium to afford the corresponding dianion, which could react directly with PCl_3_ or could be protonated first with triethylamine hydrochloride, and after that, react with PCl_3_ to afford the respective *P*-chloro-1,3,2-diazaphospholene derivatives **273**. An alternative approach affords 2-bromo-1,3,2-diazaphospholene derivatives directly by the reaction of the diazadiene **272** with PBr_3_ and cyclohexene [122]. The 2-halogenate-DAPs may further react with suitable nucleophiles under halide displacement to give products with a variety of functional *P*-substituents **274** (Figure 65), (e.g., OTf [123], OMe [124], oxide [125], H [126], azide [127], halogen [128])

#### 5.2.5. Synthesis of Dihydropyrrolo[1,2-a]pyrazine Derivatives

When the reaction is carried out in a mixture of methanol and water as the solvent, the cyclic 1,4-diazadiene (2,3-dihydro-5-methyl-6-phenylpyrazine) **223**, formed in situ by the reaction of the diamine **275** and the 1-phenyl-1,2-propanedione **276**, yields 3,4-dihydropyrrolo[1,2-a]pyrazine **277** and 3,7-dihydropyrrolo[1,2-a]pyrazin-6(4*H*)-one **278** via dehydration–condensation of **223** with the dicetone **276** (Figure 66) [129]. Theoretical calculations show that the reaction occurred via **279** through an aza–ene reaction with the carbonyl of the benzoyl group (Ph-C=O) of **276** followed by dehydration to yield **280**. The subsequent cyclization of **280** proceeds through a methyl rearrangement, yielding **278**. In parallel, the formation of **277** would be explained by the reduction of **280** using **223** as a reducing reagent, followed by cyclization and subsequent dehydration.

#### 5.2.6. Synthesis of *N*,*N*-Disubstituted Exo-2-imidazolidinone Dienes

The reaction of 1,4-diaza-1,3-butadiene **281** with triphosgene in the presence of Et_3_N yields *N*,*N*-disubstituted exo-2-imidazolidinone dienes **282** [130]. When **281** is not symmetric, the diastereoselectivity (*E*/*Z*) in the synthesis of **282** is temperature-dependent, with the *E* isomer being the main product at low temperatures (−10 °C) but increasing the reaction temperature to 20 °C yielded only the heterocyclic inner–outer ring diene **283**, which can also be obtained in good yields after treatment of **282** with AlCl_3_. Dienes (*E*)-**282** and (*Z*)-**282** are highly reactive and regioselective in Diels–Alder cycloadditions with acrolein, yielding the corresponding *ortho* adducts **284** as the major regioisomer, and the reaction with DDQ promoted aromatization of those cycloadducts, yielding the respective benzimidazol-2-ones **285** in high yield. Similarly, the reaction of dienes **283** with *N*-phenyl-maleimide was very diastereoselective, giving *endo* adducts **286** exclusively (Figure 67).

#### 5.2.7. Synthesis of Pyrrolo[3,2-b]pyrrole-1,4-dione (isoDPP) Derivatives

To obtain new materials with optoelectronic applications such as organic photovoltaics (OPVs) solar cells, organic light-emitting field-effect transistors (LEFETs), or organic light-emitting diodes (OLEDs), different types of compounds have been studied, including 1,3,4,6-tetraarylpyrrolo[3,2-b]pyrrole-2,5-dione **287** (isoDPP), which have been used in the preparation of non-polymeric and polymeric materials for optoelectronic applications [131]. One of the strategies to synthesize these compounds consists of the reaction of bis-aryloxalimidoylchlorides **288** with two equivalents of an ester enolate (Figure 68) [132]. 

The typical method for producing isoDPP derivatives involves first preparing 1,3,4,6-tetrasubstituted pyrrolo[3,2-*b*]pyrrole-2,5(1*H*,4*H*)-dione **287** as the central building block by reacting 1,4-diaza-1,3-butadiene **288** with a suitable ester enolate, such as ethyl-2-phenylacetate or ethyl-2-(thiophen-2-yl)acetate [133]. In the next step, the molecule could be modified through coupling reactions on the thiophene rings at positions 3 and 6. These modifications can introduce motifs that change the molecule’s properties and/or generate the possibility of obtaining polymeric chains. NAI-IsoDPP-NAI and PI-IsoDPP-PI were synthesized by Suzuki coupling reactions to introduce napthalimide or phthalimide as end-capping groups [134]. By using a Stille coupling reaction with organotin derivatives of thiophene, it was possible to introduce a different number of thiophene rings in the backbone [135]. However, using distannylated donor comonomers in the Stille coupling reaction, it is possible to obtain isoDPP polymers [136]. Polymers can also be synthesized via palladium-catalyzed direct C–H arylation (Figure 69) [137].

### 5.3. Six-Membered Rings

#### 5.3.1. Synthesis of 1,4-Diaza-2,3-diborinines

The reaction of the dilithium salts of 1,4-diaza-1,3-butadienes **289** with 1,2-dichlorodiborane derivatives **290** yielded 1,4,diaza-2,3-diborinines (DADB) **291** (Figure 70) [138]. These 2,3-diborinines are cyclic (amino)diboranes [4] having a cyclic structure similar to benzene with a C=C bond substituted by B-N moieties. So far, little is known about the chemistry of DADBs, and just a few examples have been reported. When 1,2-dichloro-1,2-bis(dimethylamino)-diborane(4) is used in the reaction with **289**, the yield of 1,4,diaza-2,3-diborinines **292** is much higher than when dibromo analogs are used. The reaction of **292** with BH_3_SMe_2_ at room temperature produces 1,2-dihydrodiborane(4) derivatives **293** [139]. By reacting with HX or commuting with BX_3_, the unreactive NMe_2_ groups of **292** can be readily substituted by halides [140]. When 1,2-dihydrodiboranes(4) **294** in benzene reacts with an excess of trimethylsilyl azide, 1,2-diazidodiboranes(4) **295** are cleanly obtained. Some of the derivatives are stable enough to pyrolyze in a controlled manner without explosive decomposition. Various diazadiboretidins **296** were produced as a result of this pyrolysis. These novel compounds appear to be the dimerization products of transitory, endocyclic iminoboranes [141].

#### 5.3.2. Synthesis of Dihydroquinoxaline Derivatives

Dihydroxyquinoxaline derivative **297** could be synthesized in good to excellent yields by an NHC-catalyzed α-carbon amination of aromatic or aliphatic α-chloroaldehyde **298** with cyclohexadiene-1,2-diimine **299**. This reaction could be consider an aza-[2+4] cycloaddition reaction between **298** and 1,4-diaza-1,3-butadienes **299** (Figure 71). The β-phenyl ring of the α-chloroaldehyde can have both electron-withdrawing and electron-donating groups without negatively effecting the reaction yield or its enantioselectivity. The same is observed when the cyclohexadiene-1,2-diimine **299** has electron-donating groups; however, with electron-withdrawing groups, the product is obtained in lower yields. Saturated aldehydes **300** could also be used in this reaction under oxidative NHC catalysis, but the yield is low despite the high enantioselectivity [142].

The cyclohexadiene-1,2-diimine **299** could be prepared in situ to afford the one-pot reaction (Figure 72). Oxidation of the corresponding diamine with Pb(OAc)_4_ gives the diimine **299** and the cycloaddition products **301** are obtained with moderate yield and high enantioselectivity under the preceding conditions.

An asymmetric catalytic inverse electron demand hetero-Diels–Alder reaction of ketene enolates and ortho-benzoquinone diimides **302**, catalyzed with benzoylquinidine **303** and Zn(OTf)_2_, can yield similar quinoxalinone with high enantioselectivity. Hünig’s base and **303** react with the acyl chloride **304** to produce in situ the respective ketene enolates, which then react with **302** to yield quinoxalinone derivatives **305** in one step. A stepwise process is consistent with the observed regiochemistry of quinoxalinone derivatives **305** obtained (Figure 73) [143].

## 6. Conclusions

Diaza-1,3-butedienes are versatile building blocks that can construct fussed and single heterocyclic compounds with four-, five-, and six-membered rings. The approaches presented in this review provide a synthetic tool for constructing alkaloid cores, *N*-heterocyclic carbenes, and other bioactive molecules. Isolated and in situ diaza-1,3-butadienes, produced from their respective precursors (typically imines and hydrazones) under a variety of conditions, can both be used to make these heterocyclic compounds. Cycloadditions, Diels–Alder, inverse electron demand Diels–Alder, and aza-Diels–Alder reactions of 1,2-diaza-, 1,3-diaza-, and 1,4-diaza-1,3-butadienes with a variety of substrates allows access to complex structures via C-C, C-N, and C-S bond formation in ring-closing procedures, while avoiding the use of expensive transition metal organometallic compounds. Nucleophilic additions and Michael-type reactions to 1,2-diaza-, 1,3-diaza-, and 2,3-diaza-1,3-butadienes can provide reactive intermediates that can be cyclized to produce heterocyclic cores. We hope that this review serves as an update for synthetic chemists and that the new insights gained here will lead to new techniques and methods for synthesizing new bioactive heterocycles.

## Data Availability

All tables are created by the authors. All sources of information are adequately referenced. There is no need to obtain copyright permissions.

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
