# Peer review of "Diaza-1,3-butadienes as Useful Intermediate in Heterocycles Synthesis"

_molecules, 2022, doi:10.3390/molecules27196708_

Round 1
Reviewer 1 Report
The manuscript by Heredia-Moya et al summarized applications of diaza-1,3-butadienes for the syntheses of functional heterocyclics. The employments of 1,2-diaza-, 1,3-diaza-, 2,3-diaza, and 1,4-diaza-1,3-butadienes as intermediates for synthesizing heterocycles during the last decade were well summarized in this review, which involves nucleophilic addition, Michael reaction, and a variety of cycloadditions. The paper is well written overall and would be a useful summary for readers in the related area. In this respect, the paper could be accepted for publication in molecules.
Only one suggestion that I have: it would be useful to draw a scheme for the synthetic methods for the preparation of diaza-1,3-butadienes (may combine with Scheme 1), although some of them were already described in the text.
Reviewer 2 Report
The manuscript "Diaza-1,3-butadienes as useful intermediate in heterocyclic synthesis" by Moya and co-workers was checked carefully and found to be suitable for publication.
It is a comprehensive review that nicely summarizes the chemistry of diaza-1,3-butadienes, particularly for the synthesis of various heterocyclic compounds. The title should be "Diaza-1,3-butadienes as useful intermediate in heterocycles synthesis" or "Diaza-1,3-butadienes as useful intermediate in heterocyclic compounds synthesis". The term "heterocyclic synthesis" does not make sense. There are some typographical errors in the manuscript which need to be corrected before publication. 1. Scheme 4, "Reflux" should be "reflux". 2. Scheme 4, "Toluene" should be "toluene". 3. Schemes 30, and 32 also contain typos. 4. Sheme 34, "X, Y, Z = CN, N" should be "X, Y, Z = CH, N". same mistake in "Y = CN, N." 5. Lots of typographical errors need to be carefully corrected. 6. In isoquinoline synthesis, I will recommend the following three papers where oxygen-containing hetero-diene reacts with dienophile to produce isoquinolines through the (4+2) Benzannulation process. (a) Synthesis 2013, 45(9): 1227-1234; (b) Synlett 2011, 05, 0689; (c) Beilstein Journal of Organic Chemistry 2009, 5, No. 35. 7. Some references, like reference 27, have typographical errors. Thus, with the abovementioned corrections, I am recommending this manuscript for publication in Molecules.Author Response
Please see the attachment.

Reviewer 3 Report
In this article, Heredia-Moya et al present a review of the use of 1,2-diaza-, 1,3-diaza-, 2,3-diaza-, and 1,4-diaza-1,3-butadienes as intermediates to synthesize N-heterocyclic compounds. Overall, this review is well written and of great interest to the readers. I recommend publication of this manuscript after minor revision.
1. The yields of the N-heterocyclic compound should be added.
2. A list of abbreviation should be added to help reader.
